# Momentum-resolved fingerprint of Mottness in layer-dimerized $Nb_3Br_8$

Mihir Date [1,2], Francesco Petocchi[3], Yun Yen[4,5], Jonas A. Krieger [1,6], Banabir Pal [1], Vicky Hasse[7], Emily C. McFarlane [1], Chris Körner [8], Jiho Yoon [1], Matthew D. Watson [2], Vladimir N. Strocov [9], Yuanfeng Xu[10], Ilya Kostanovski [1], Mazhar N. Ali[1,11], Sailong Ju[9], Nicholas C. Plumb [9], Michael A. Sentef [12,13], Georg Woltersdorf [8], Michael Schüler [4,14], Philipp Werner [14], Claudia Felser [7], Stuart S. P. Parkin [1] & Niels B. M. Schröter [1] ✉

Crystalline solids can become band insulators due to fully filled bands, or Mott insulators due to strong electronic correlations. While Mott insulators can theoretically occur in systems with an even number of electrons per unit cell, distinguishing them from band insulators experimentally has remained a longstanding challenge. In this work, we present a unique momentum-resolved signature of a dimerized Mott-insulating phase in the experimental spectral function of $Nb_3Br_8$: the top of the highest occupied band along the out-of-plane direction $k_z$ has a momentum-space separation $\Delta k_z = 2\pi/d$, whereas that of a band insulator is less than $\pi/d$, where d is the average interlayer spacing. Identifying $Nb_3Br_8$ as a Mott insulator is crucial to understand its role in the field-free Josephson diode effect. Moreover, our method could be extended to other van der Waals systems where tuning interlayer coupling and Coulomb interactions can drive a band- to Mott-insulating transition.

Whilst the single-electron Schrödinger equation can describe most ordinary metals, crystalline insulators generally fall into two classes: band insulators, which form band gaps due to the interaction of electron waves with a periodic potential which the single-electron Schrödinger equation can describe; and strongly correlated insulators, such as Mott insulators, in which the interaction between electrons leads to the formation of a gap that cannot be fully explained in the single-electron picture. Conventional Mott-insulators are typically found in materials that should be metals according to band theory due to an odd number of electrons per unit cell, but turn out to be insulators due to strong electron-electron interactions. For an even number of electrons, however, both Mott- and band-insulating phases can theoretically occur. Distinguishing such an unconventional Mott-insulator from a band insulator is crucial because the Mott-phase is

[1]Max Planck Institut für Mikrostrukturphysik, Weinberg 2, 06120 Halle, Germany. [2]Diamond Light Source Ltd, Harwell Science and Innovation Campus, Didcot OX11 0DE, UK. [3]Department of Quantum Matter Physics, University of Geneva, 24 Quai Ernest-Ansermet, 1211 Geneva 4, Switzerland. [4]PSI Center for Scientific Computing, Theory and Data, Paul Scherrer Institute, CH-5232 Villigen PSI, Switzerland. [5]École Polytechnique Fédérale de Lausanne (EPFL), CH-1015 Lausanne, Switzerland. [6]PSI Center for Neutron and Muon Sciences CNM, 5232 Villigen PSI, Switzerland. [7]Max Planck Institute for Chemical Physics of Solids Nöthnitzer Straße, 40 01187 Dresden, Germany. [8]Martin-Luther-Universität Halle-Wittenberg, Von-Danckelmann-Platz 3, 06120 Halle (Saale), Germany. [9]Swiss Light Source, Paul Scherrer Institute, CH-5232 Villigen PSI, Switzerland. [10]Center for Correlated Matter and School of Physics, Zhejiang University, Hangzhou 310058, China. [11]Kavli Institute of Nanoscience, Delft University of Technology, 2628 CJ Delft, The Netherlands. [12]Institute for Theoretical Physics and Bremen Center for Computational Materials Science, University of Bremen, 28359 Bremen, Germany. [13]Max Planck Institute for the Structure and Dynamics of Matter, Center for Free-Electron Laser Science (CFEL), Luruper Chaussee 149, 22761 Hamburg, Germany. [14]Department of Physics, University of Fribourg, 1700 Fribourg, Switzerland. ✉e-mail: niels.schroeter@mpi-halle.mpg.de

the parent phase for many strongly correlated phenomena, including exotic magnetism and superconductivity. This distinction has inspired extensive theoretical studies of the band- to Mott-insulator transition[1–15], where Mott-insulators are often identified by a divergence in the self-energy[16,17].

Experimentally identifying an unconventional Mott-insulator with an even number of electrons is difficult, though, since both band- and Mott-insulators exhibit energy gaps in their excitation spectra, which lead to similar signatures in optical and transport measurements. Whilst some attempts have been made to identify such unconventional Mott-insulating phases via energy-resolved spectroscopy[18–20], interpreting the results can be challenging because the density of states near the Fermi level can be influenced by many factors, such as disorder or many-body interactions. Since these experiments lacked momentum resolution, an unambiguous momentum-space fingerprint of unconventional Mott insulators has remained elusive in experiments. These challenges have led to longstanding debates about the true nature of the insulating ground state in many important quantum materials, such as $1T$-$TaS_2$[18–24], $VO_2$[25–28], and $CoO$[29–33]. Finding an unambiguous experimental signature in momentum space that could identify Mott-insulators is therefore a critical unsolved problem in condensed matter physics.

This challenge has recently gained significance for the van der Waals material $Nb_3Br_8$, which has previously been assumed to be a band insulator in its low-temperature layer-dimerized form[34–36]. When a few layers of dimerized $Nb_3Br_8$ are sandwiched between superconducting $NbSe_2$ electrodes, the Cooper-pair tunneling across the $Nb_3Br_8$ weak link has been reported to show a non-reciprocal critical current in the absence of a magnetic field, known as the field-free Josephson diode effect (JDE)[37]. Such a magnetic field-free JDE is highly unexpected because Onsager relations predict that the JDE would need to break time-reversal symmetry[38], but magnetic susceptibility and muon spectroscopy point towards a non-magnetic ground state of $Nb_3Br_8$[39,40]. It has been speculated that so-called obstructed surface states that are located in every other van der Waals (vdW) gap of the assumed layer-dimerized band insulator $Nb_3Br_8$[34,41] could create an out-of-plane polarization that may induce asymmetric Josephson tunneling[37]. However, so far, no experimental evidence for such obstructed surface states has been reported. If, on the other hand, $Nb_3Br_8$ was an unconventional Mott insulator, it has been predicted that hole-doping a monolayer could create a time-reversal symmetry-breaking topological superconductor[35], a state that may help to explain the field free diode effect. It is conceivable that at the interface between $Nb_3Br_8$ and the metallic $NbSe_2$ a charge transfer takes place that could realize a locally doped, two-dimensional $Nb_3Br_8$ layer, which could show very different superconducting behavior depending on the band or Mott insulating nature of the $Nb_3Br_8$ parent state[38]. It is therefore important to identify an experimental signature that could distinguish between band and Mott insulating behavior in layer-dimerized $Nb_3Br_8$, as this could help us understand the mysterious JDE reported in heterostructures and also inspire further experiments in other strongly correlated materials where the band- vs. Mott-insulator distinction remains hotly debated. In this work, we propose such a distinguishing fingerprint in momentum space and demonstrate that layer dimerized $Nb_3Br_8$ is inside the Mott-phase.

To understand the competition between band- and Mott-insulating phases in $Nb_3Br_8$, it is instructive to inspect its crystal structure as shown in Fig. 1a. In a single layer, Nb atoms form an in-plane trimer in the so-called breathing Kagome lattice, which results in a $2a_1$ molecular orbital localized over each $[Nb_3]^{8+}$ cluster[39]. In its low temperature form, such trimers are stacked in two distinct ways along the out-of-plane direction—first, in which the trimers from adjacent layers are exactly atop each other (Fig. 1b) and second, in which they are shifted (Fig. 1c). In Fig. 1b, the layers are coupled and are closer to one another compared to the ones in Fig. 1c. The crystal structure

therefore consists of dimerized bilayers that are weakly coupled, classified with space group R$\bar{3}$m #166. The non-equivalence of these bilayer stacks is reflected in the hopping constants, which are large for intra-dimer hopping ($t_1$), and small for inter-dimer hopping ($t_2$), as shown in Fig. 1d.

Theoretical studies of $Nb_3X_8$ (X = Cl, Br, I) monolayers predicted that these materials host half-filled bands with small bandwidth that become unstable against electronic interactions and form Mott insulators with antiferromagnetic ground states[35,42]. When these monolayers are stacked into bilayers, ref. 35 finds a crossover between a band insulator and Mott insulator depending on the interlayer hopping and correlation strength, with bilayers of $Nb_3Br_8$ predicted to lie in the band insulating regime. In contrast, the sister compound $Nb_3Cl_8$ is not considered to be a band insulator but a Mott insulator due to the weaker interlayer coupling[35,42–44]. These theoretical results are very instructive for understanding the possible electronic phases of few-layer samples of $Nb_3Br_8$ that have been studied in the field free JDE. Depending on the strength of intra-bilayer and inter-bilayer hopping, as well as the effective magnitude $U$ of electron-electron interactions, one can expect that the few-layer and bulk $Nb_3X_8$ could potentially host either a band-insulating phase, where the gap stems from layer-dimerization, or a Mott-insulating phase, where electrons are localized by correlations, but with fingerprints of dimerization still present. A hypothetical phase diagram for bulk $Nb_3Br_8$ is illustrated in Fig. 1e, inspired by the calculated diagram for bilayer $Nb_3X_8$ (X = Cl, Br) in ref. 35: the metallic and insulating ground states are accessible by tuning the parameter $\eta = |t_1 - t_2|$, which indicates the degree of layer-dimerization, and the onsite interaction strength $U$. Given the sizeable degree of layer-dimerization and high resistance of $Nb_3Br_8$, we focus on the insulating phases, particularly the dimerized band and dimerized Mott insulators, highlighted in Fig. 1e. Theoretically it is possible to identify the transition from a dimerized band-insulator toward a dimerized Mott-insulator by following the evolution of the local self-energy $\Sigma$ as a function of the interaction strength, as shown in Fig. 1f. In the weak coupling regime $\Sigma$ vanishes at low-energy and results in a marginal renormalization of the bare bandstructure. By increasing $U$ the self-energy develops a divergence, pinned to the Fermi level in the case of particle-hole symmetric systems, which is the hallmark of the Mott insulating regime. An intriguing scenario arises when the band-structure already hosts a hybridization gap due to interlayer coupling in the absence of strong electron correlations: in such a case the mere presence of a gap in the spectra is not enough to infer the microscopic nature of the insulating state. Moreover, since the self-energy is not directly observable, it is a formidable task to distinguish a dimerized band-insulator from a dimerized-Mott insulator experimentally.

Here, we propose a measurable signature that differs for the two insulating cases: the crystal momentum position of the top of the highest-occupied band of the one-electron spectral function along the dimerization direction. For the case of $Nb_3Br_8$, this is the out-of-plane $k_z$-axis orthogonal to the layers. To the best of our knowledge, this signature has not been experimentally investigated in $Nb_3Br_8$ or any other correlated layered insulator. To gain an intuitive understanding of this signature, one can start by considering the spectral function of a half-filled metallic fermi-liquid with lattice constant $a$ along the momentum direction $k_z$, as displayed Fig. 1g. The unoccupied top of the conduction band is located at $k_z = \pi/a$. When considering a band insulator created by dimerization along the direction of $k_z$, it will have a valence band maximum in the spectral function at $k_z = \pi/2a$, as shown in Fig. 1h. The gap opening can be understood as a band hybridization between the original and backfolded bands due to the doubling of the unit cell leading to an even number of electrons. For relatively weak dimerization, the backfolded bands are expected to be weak in intensity as well. As long as the on-site correlations are sufficiently weak, the valence band maximum will remain at $k_z = \pi/2a$ even if the dimerization gap is renormalized. In contrast, an undimerized Mott

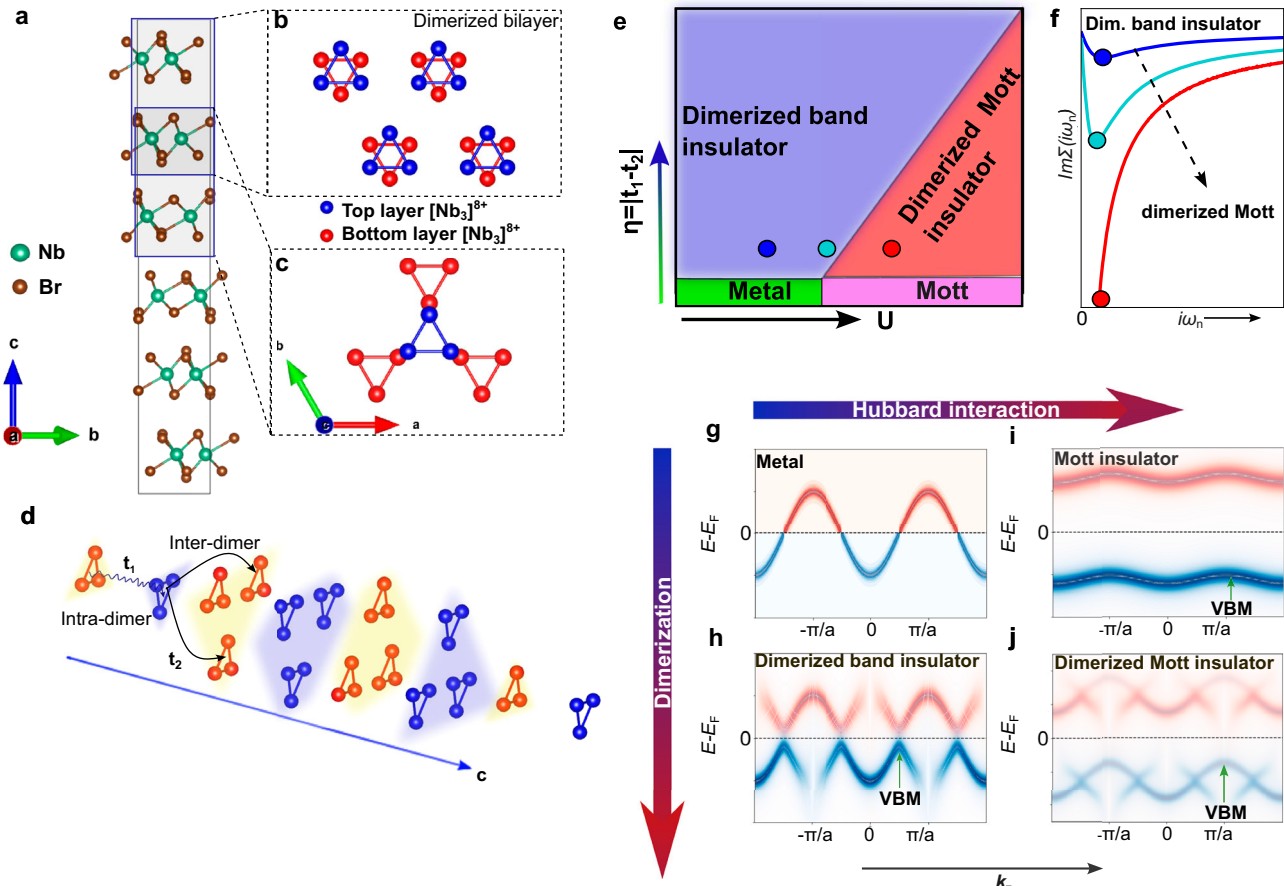

**Fig. 1 | Crystal structure and dimerized band- to Mott-insulator transition in Nb$_3$Br$_8$. a** The conventional unit cell of layer-dimerized Nb$_3$Br$_8$. Layers of Nb$_3$Br$_8$ are stacked alternately in the configuration shown in (**b**) and (**c**). **d** Interlayer electron hopping across these stacks effectively depicted as a 1D chain of half-filled 2a$_1$ dimers. **e** A schematic depiction of different phases realizable for Nb$_3$Br$_8$, where $\eta = |t_1 - t_2|$ describes the dimerization strength and U the on-site Hubbard interaction. **f** Evolution of imaginary part of self-energy when increasing U across the dimerized band- to Mott-insulator transition. $\omega_n$ is the Matsubara frequency. The color of the curves indicate different strengths of U indicated by the colored dots in the phase diagram in (**e**). **g–j** Illustrated dispersion of spectral function for three-dimensional solid that undergoes dimerization along the momentum direction $k_z$. Here, $a$ is the lattice constant of the metal before dimerization. The occupied spectrum shaded in blue is detected in the ARPES experiment, whereas the unoccupied one in red is not.

insulator with on-site interactions significantly larger than the bandwidth will show the maximum of the lower Hubbard band at $k_z = \pi/a$ as illustrated in Fig. 1i. When the dimerization is relatively weak, but the on-site interactions are strong, the system will form a dimerized Mott insulator with a spectral function that has a pronounced maximum at $k_z = \pi/a$, and backfolding leads to a dimerization gap at $k_z = \pi/2a$ between the original Hubbard bands from the undimerized phase, and the backfolded Hubbard bands due to the doubling of the unit cell. The backfolded Hubbard band is also expected to show weak spectral weight for a relatively weak dimerization. In that case, the separation between the top of the highest occupied band is $\Delta k_z = \pi/a$ for the dimerized band insulator, whilst it is $\Delta k_z = 2\pi/a$ for the dimerized Mott insulator. For this purpose, we define a *band* as continuous feature in the spectral function with significant spectral weight, dispersing along a specific $k$–path.

In this work, we analyze the experimental spectral function to show that Nb$_3$Br$_8$ is in the strongly correlated dimerized Mott insulating phase. To support this claim, we present detailed ARPES spectra of the in-plane and out-of-plane electronic structure of Nb$_3$Br$_8$, as well as infrared absorption spectroscopy to determine the size of the optical spectral gap at the Fermi-level in Nb$_3$Br$_8$. We find that the in-plane spectral function of Nb$_3$Br$_8$ is very similar to the electronic structure predicted by DFT calculations. This similarity explains why previous ARPES studies have *not* identified Nb$_3$Br$_8$ as a Mott insulator,

but as a semiconductor[36]. The measurement of the out-of-plane spectral function, however, clearly reveals the Mott-insulating character of dimerized Nb$_3$Br$_8$ despite sizable interlayer coupling. This interpretation of our data is further supported by simulations based on dynamical mean field theory (DMFT).

## Results and discussion

Before presenting a fingerprint of the Mott-insulating state in the out-of-plane dispersion of the spectral function, we will discuss its in-plane dispersion and the optical band gap. The three-dimensional Brillouin zone of Nb$_3$Br$_8$, and its surface projection (see Supplementary Information for convention) is shown in Fig. 2a and the results of our infrared-absorption measurements are shown in Fig. 2b. The extracted optical band gap of Nb$_3$Br$_8$ is 370 meV, which is much larger than a direct gap at the T-point of around 60 meV predicted by our DFT calculations. This discrepancy might arise due to correlations that renormalize the size of the band gap. However, the gap renormalization does not conclusively differentiate between band- and Mott-insulator. We show calculated DFT band structures along the Γ-L and T-M directions in Fig. 2c, d, respectively, which are contrasted with ARPES spectra measured at two different photon energies, $h\nu = 85$ eV shown in Fig. 2e, g, and $h\nu = 67$ eV shown in Fig. 2f, h. We have chosen these two energies for comparison because the spectral peaks of the band maximum at normal emission ($\bar{\Gamma}$ point) shown in the line-cuts of

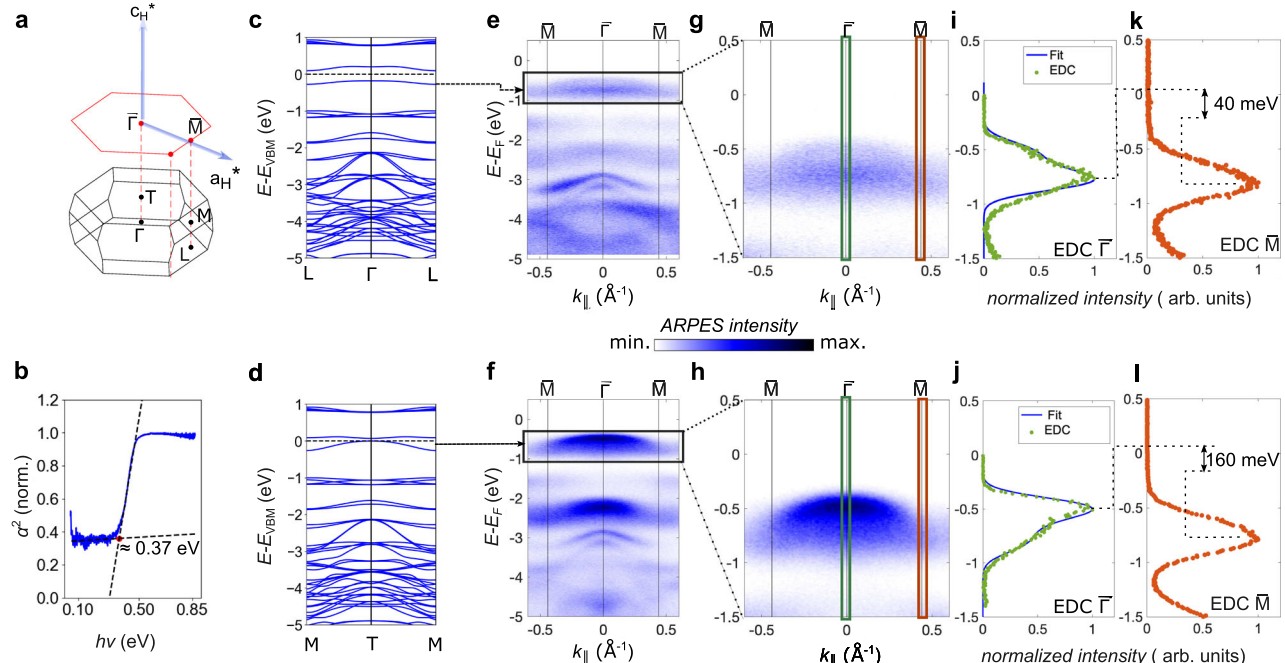

**Fig. 2 | In-plane dispersion of spectral function. a** Three-dimensional and projected Brillouin zone of $Nb_3Br_8$. In the projected surface Brillouin zone (red hexagon), the reciprocal lattice vectors $a_H^*$ and $c_H^*$ are represented in the hexagonal coordinate system. **b** Room temperature infrared absorption spectrum of $Nb_3Br_8$ shown as a Tauc plot. The optical bandgap is extracted from the intersection of the leading edge with the baseline. **c, d** show DFT bandstructures in the $\Gamma$- and T-planes, respectively. The in-plane electronic structure of $Nb_3Br_8$ in the planes **e** $k_z = 0$ measured at $h\nu = 85$ eV and **f** $k_z = \pi$ measured at $h\nu = 67$ eV. **g, h** show zoomed-in valence band maxima in the two $k_z$ planes. The green and red-orange windows show integration windows for EDCs in (**i**) and (**j**). The blue curve in **i** and **j** are a fit that includes $k_z$ broadening (model described in Supplementary Information). **k, l** illustrate the in-plane bandwidth ($\delta_{BW}$) of the VBM in the two $k_z$-planes, respectively, as extracted from the peaks of the EDCs (indicated by the dashed lines) at the $\bar{\Gamma}$, and $\bar{M}$ point.

Fig. 2i, j are at the minimal and maximal binding energy, respectively, as would be expected for the $\Gamma$ and T point in the respective DFT calculations. Except for the underestimated band gaps between the bands, the DFT calculations describe the experimental in-plane dispersions qualitatively rather well. We note that a clear shoulder is visible in the energy distribution curves (EDCs) in Fig. 2i, j, which could either originate from a Hubbard band split due to the layer dimerization, or from the so-called $k_z$ broadening of the valence band of a band insulator due to the finite escape depth of the photoelectrons[45]. We perform a fit to the EDCs, indicated by the blue curve in Fig. 2i, j, where we express the intensity as a convolution of energy broadening and $k_z$ momentum broadening, and find reasonable agreement with the experimental data with slight deviations for the curve in Fig. 2i (see Supplementary Information for details of the fitting model). The observed shoulder can therefore be explained by both band- or Mott-insulator phases. We can estimate the bandwidth along the in-plane direction from the peak shift of the EDC between the $\bar{\Gamma}$ and $\bar{M}$ points, which varies between 40 meV (Fig. 2k) and 160 meV (Fig. 2l), respectively, depending on the respective high-symmetry plane that is probed along the $k_z$ direction.

To clearly distinguish the Mott- from the band-insulator, one needs to carefully investigate the spectral function along the dimerization direction, which is the out of plane direction $k_z$ in $Nb_3Br_8$, and compare it with the theoretical expectations illustrated in Fig. 1h–j.

In Fig. 3a, we plot the spectral function along the $k_z$ direction obtained performing a photon energy dependent ARPES measurement over an energy range of 55 eV to 120 eV. We find that the top of the highest occupied band is separated by approximately $\Delta k_z = 0.89\,\text{Å}^{-1}$ in momentum space, approximately twice the distance of T-$\Gamma$-T in the primitive Brillouin zone (cf. Fig. 2a), which corresponds to $k_z = 0.44\,\text{Å}^{-1}$. Intriguingly, the observed periodicity of the

highest occupied band is numerically equal to $2\pi/d = 0.89\,\text{Å}^{-1}$, where $d$ is the average spacing between Nb planes. Furthermore, noting that the interlayer spacing in alternate vdW gaps only differs by $0.18\,\text{Å}$, the dimerization can be considered only a small perturbation to a hypothetical undimerized parent structure of equally spaced and stacked layers with out-of-plane lattice constant $d$. These observations provide concrete evidence that the layer dimerized $Nb_3Br_8$ is in the Mott phase because the maxima of the Hubbard bands in a dimerized Mott insulator are expected to have a periodicity that corresponds to the reciprocal lattice constant of the undimerized Brillouin zone ($\Delta k_z = 2\pi/d$ cf. Fig. 1j), whilst the dimerized band insulator is expected to have a periodicity with half of that lattice constant (cf. Fig. 1h). The position of the maxima and minima of the bands can also be extracted from line cuts, so-called momentum distribution curves (MDCs), shown in Fig. 3e. The blue scatter plot shows MDC intensity peaks, integrated in the energy window around the band maximum, spaced by $0.89\,\text{Å}^{-1}$. On the other hand, we notice that in the red scatter plot corresponding to the MDC of the band minima show an anomalous peak at $\approx 4.3\,\text{Å}^{-1}$, which does not seem to obey this periodicity. While we do not have a conclusive explanation for this discrepancy, we point out that such behavior is only observed for lower photon energies in the VUV range, where the final state may not be free-electron like. This can lead to deviations in the observed photoemission spectra from the expected one. Nevertheless, the observed ARPES dispersion in Fig. 3a agrees qualitatively well with the predicted dispersion for a dimerized Mott insulator in Fig. 3c. Furthermore, the $2\pi/d$ periodicity of the band maxima is also observed over a wider momentum range when probed with soft X-rays (see Supplementary Fig. 4), where the final state can be expected to become more free-electron like. Therefore, we conclude that the overall data strongly points towards the dimerized Mott insulating phase in $Nb_3Br_8$. The average hopping

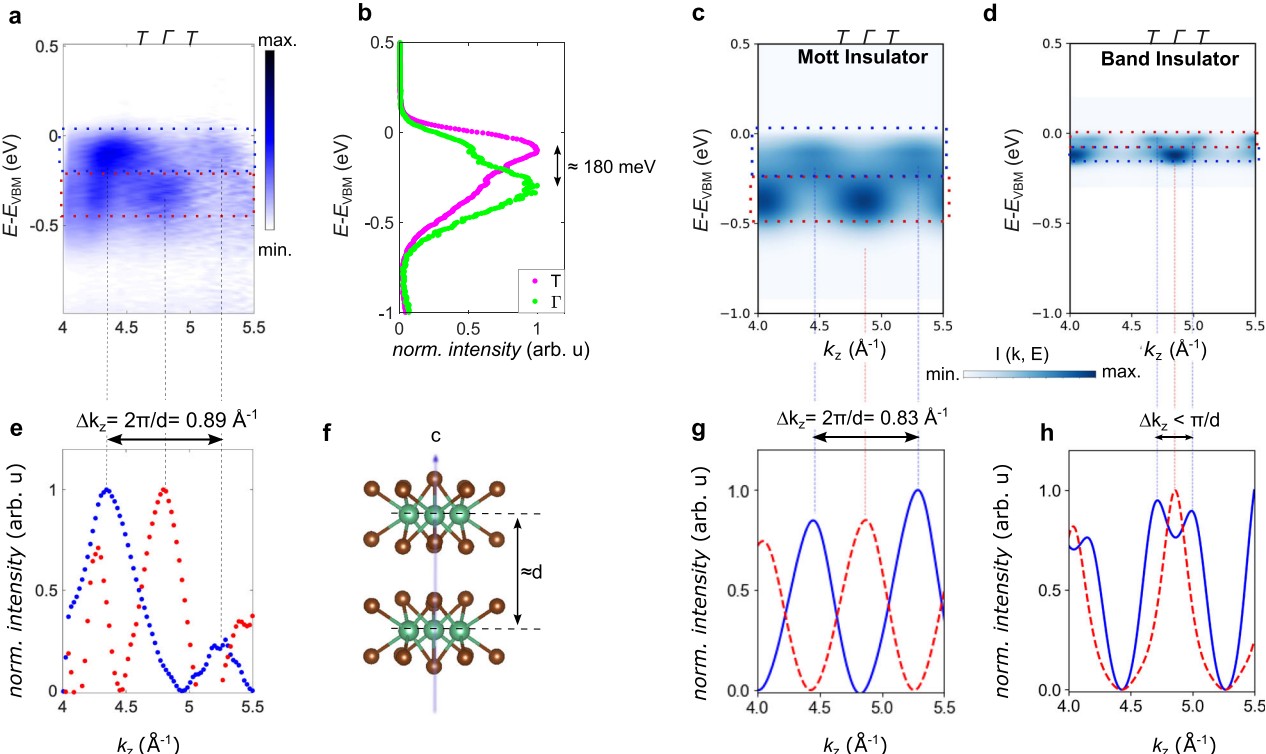

**Fig. 3 | Out-of-plane dispersion of the spectral function of the highest occupied band. a** ARPES spectrum at normal emission measured with linear horizontal polarization as a function of out-of-plane momentum $k_z$, converted from photon energy dependence via the free electron final state approximation with inner potential $V_0 = 12$ eV. **b** EDCs at the band top and bottom showing the bandwidth of the $k_z$ dispersion. **c**, **d** show ARPES simulations of the effective 1D dimerized system in the Mott insulating phase, and in the band insulating phase, respectively. The corresponding intensity profiles $I(k,E)$ along $k_z$ are shown in (**g**) and (**h**), which we use to compare with the MDCs of (**a**), integrated over the red and blue energy windows, displayed in (**e**). **f** The interlayer spacing $d$ corresponds to the average distance between the planes of Nb atoms.

along the out-of-plane direction can be estimated from the energy difference between the minimum and maximum of the band, which is about 180 meV, as can be seen from the energy distribution curves, shown in Fig. 3b. To further support the identification of the Mott phase, we performed simulations of the ARPES spectra for the experimental geometry, with an input model from dynamical mean field theory (DMFT). Given that the out-of-plane dispersion is predominantly described by the half-filled $2a_1$ states formed by Nb-clusters (cf. Supplementary Fig. 2), we restricted the effective model to a single band, formed by a vertically stacked triangular lattice planes. Guided by the experimental bandwidths, the hopping constants were chosen to describe the essential features of the in-plane and out-of-plane dispersions adequately. With the interaction strength of $U = 0.86$ eV the DMFT solution qualitatively reproduces the optical gap and provides significant spectral weight at $k_z = \pm \frac{\pi}{d}$, thus reproducing both the experimental band dispersion (cf. Fig. 3c) and the experimental MDCs (cf. Fig. 1j). With these parameters, we found the local self-energy to be diverging at low frequency, thus placing the system in the Mott insulating regime (c.f. Fig. S3 for details). For comparison, we also show a simulated spectrum of a band insulator in Fig. 3d, which was computed with the same hopping parameters but vanishing on-site interaction, and clearly does not reproduce the experimentally measured dispersion. We find the spacing between the intensity peaks of the valence band maximum in the MDC shown in Fig. 3h to be less than $\Delta k = \pi/d$, which is much smaller than in the experiment and therefore clearly indicates the presence of the Mott transition. For further details about the ARPES simulations, see the Supplementary Information.

Finally, we address the absence of the predicted metallic obstructed surface states in our experimental spectra of $Nb_3Br_8$. The most probable explanation seems to be that most of the cleaved surface has a termination that is created by breaking the bonds between the weakly coupled bilayers. As a result, no obstructed surface states are expected to occur because there are no unpassivated dangling bonds left at the surface. An alternative explanation is that the obstructed surface states do exist, but they are not metallic because they are gapped out by interactions. To conclude, we presented an approach to distinguish band- and Mott-insulating behavior in $Nb_3Br_8$ based on the dispersion of the spectral function along the out-of-plane direction. We find the momentum spacing between the maxima of the highest occupied band is approximately $2\pi/d$, which means that $Nb_3Br_8$ is clearly a dimerized Mott insulator, and not a band insulator as previously assumed[35,36]. This finding will motivate further investigations into the charge transfer at the $Nb_3Br_8$/$NbSe_2$ interface, which could explain the appearance of time-reversal symmetry breaking superconductivity and field-free JDE at this junction. Our approach may also be applied to other correlated insulators, such as strongly correlated heterostructures of two-dimensional materials that can be tuned both in terms of screening electronic interactions through dielectric environments and controlling interlayer interactions via deterministic control of stacking and twist angles[35]. A quantum phase transition from band- to Mott-insulator could then be identified by a spectral weight transfer from $k_z = \pi/2d$ (band insulator, cf. Fig. 1h) to $k_z = \pi/d$ (Mott insulator, cf. Fig. 1j), where $d$ is the average interlayer spacing. Interestingly, at some combination of hopping strength and onsite correlations, temperature control can even lead to coexisting band and Mott insulating behavior[15], which could potentially be explored in such tunable platforms. Identifying the control knobs that drive such a transition could in turn enable the discovery and understanding of new correlated phenomena arising from the Mott phase in

these materials, such as unconventional superconductivity or magnetism.

## Methods

### Sample growth

The single crystals of $Nb_3Br_8$ were grown by chemical vapor transport. The polycrystalline $Nb_3Br_8$ powder was synthesized by stoichiometric niobium (Nb, powder, Alfa-Aesar, 99.8%) and niobium(V)bromide ($NbBr_5$, powder, Merck, 98%) sealed in an evacuated fused silica ampule by 600 °C. Single crystals were grown by chemical vapor transport, the evacuated silica ampule heated in a two-zone-furnace between 800 °C (T2) to 760 °C (T1) for 10 days[46]. After the reaction, the ampule was removed from the furnace and quenched in water. The black-flakes crystals were characterized by powder XRD (Huber Guinier G670) and single crystal XRD (Rigaku, Saturn 724+). Exfoliated samples glued onto a carbon tape were characterized by Rutherford Backscattering Spectroscopy (NEC Pelletron) using 1.9 MV He+ beam to verify the Nb:Br stoichiometry.

### ARPES measurements

The spectra shown in Fig. 2e–h were measured at 67 eV (f and h) and 85 eV (e and g) using linear horizontal polarized vacuum UV radiation at the nano-branch of the I05 beamline[47] at the Diamond Light Source Ltd, where the approximate diameter of the beamspot (FWHM) was 4 μm. The out-of-plane dispersion displayed in Fig. 3 was measured by varying the photon energy between 55 eV and 121 eV. Our samples were measured at ≈ 235 K (to mitigate the charging effect due to the photoelectrons), at a pressure of approximately $1-2 \times 10^{-10}$ mbar, using the Scienta DA30 analyzer with a combined energy resolution of ≈ 25 meV. In Fig. 2e, f, the Fermi level offset was determined by using gold reference measurements at identical physical conditions.

The soft X-ray ARPES data presented in the Supplementary Information (Fig. S3), showing the $k_z$ dispersion in the bulk of the sample, were measured at the ADRESS beamline[48,49] of the Swiss Light Source, with right-circular polarized light, using a SPECS analyzer with an angular resolution of 0.07°. The samples were cleaved at ≈ 20 K and measured at 295K. The bulk $k_z$ dispersion was measured by varying the photon energy between 280 eV and 950 eV, which offered a combined energy resolution of 123–222 meV in the photon energy range. The position of the valence band maxima in Fig. 3a and Supplementary Fig. 3a was located by fitting the Fermi-Dirac function to the EDC.

### FTIR spectroscopy measurements

To determine the optical band gap we measured absorption spectra by means of Fourier transform infrared spectroscopy in transmission geometry (Bruker Tensor 37). The spectra obtained cover a range from 50 meV to 900 meV. In this geometry only the intensity transmitted through the sample can be measured. It is reduced by absorption inside the sample where the onset of absorption marks the band gap energy. However, the transmitted intensity also is reduced by reflection off the samples front and back surface as well as by diffuse scattering of the infrared light due to surface irregularities and roughness. We assume an energy independent reflectivity of the sample and account for the reflection by normalizing the intensity maximum of the spectra to one. Additionally, the absorption is influenced by intra-gap states which can be present due to impurities. This leads to absorption of light with photon energies below the intrinsic band gap, where the material should otherwise be perfectly transparent. By normalizing the spectra this is accounted for as well. Additionally we normalize the absorption to the thickness of the samples, which was measured by the DEKTAK profilometer at multiple positions and averaged. To recover the optical band gap from the absorption spectra we employ Tauc's method[50]. We calculate the absorption coefficient using the Beer-

Lambert law, assuming a direct band gap for the materials. In this case a plot of the square of the absorption coefficient versus photon energy reveals a step-like trace in the data. The linearly sloped section is fitted. The zero-intersection marks the onset of absorption and thus the optical band gap energy.

### DFT calculations

We used plane-wave density functional theory, as implemented in the QUANTUM ESPRESSO package[51,52] to compute the bandstructures shown in Fig. 2c, d, and Supplementary Fig. 2. The electron-ion interaction was accounted for by ultrasoft pseudopotentials compiled from the PSLIBRARY database[53], and the exchange-correlation functional was described by the revised Perdew-Burke-Erzenhoff (PBEsol) parametrization of the generalized gradient approximation (GGA)[54,55]. A semi-empirical Grimme-D3 functional[56] was used to treat van der Waals interactions, and the lattice was relaxed until the forces on each atom were less than $10^{-4}$ Ry/Bohr. The kinetic energy cut-off for the plane-waves was set to 90 Ry and a $8 \times 8 \times 6$ Monkhorst-Pack[57] k-mesh was used for Brillouin zone integration.

### DMFT simulations

The DMFT calculation employed a momentum grid of $30 \times 30 \times 30$ k-points and a high frequency cutoff of 30 eV on the Matsubara axis, for an inverse temperature of $\beta = 50$ eV$^{-1}$. Given the presence of two equivalent sites within the unit cell we solved only one impurity model using a continous-time QMC algorithm[58] and copied the local self-energy to the other lattice site at each cycle in the self-consistency loop. This approach excludes by construction any local symmetry breaking. To account for local electron-electron correlations we included an on-site Hubbard interaction $U$ representing the energy cost of doubly occupied sites, consistent with the description in Grytsiuk et al.[42]. The model Hamiltoinan is thus given by:

$$H = -t_p \sum_{\alpha \langle n,m \rangle z} c^\dagger_{\alpha,n,z} c_{\alpha,m,z} + U \sum_{\alpha n} \hat{n}_{\alpha,n,z\uparrow} \hat{n}_{\alpha,n,z\downarrow} \tag{1}$$

$$-t_D \sum_{\alpha \neq \beta n z} c^\dagger_{\alpha,n,z} c_{\beta,n,z} \tag{2}$$

$$-t_v \sum_{zn} \left[ c^\dagger_{1,n,z-1} c_{2,n,z} + c^\dagger_{2,n,z+1} c_{1,n,z} \right]. \tag{3}$$

Here, the fermionic creation (annihilation) operators $c^\dagger(c)$ are labeled by three indices: the greek one refers to the site within the unit cell ($\alpha \in \{1,2\}$), the second index labels adjacent unit cells within the plane, while the third one labels different unit cells in the stacking direction. Explicitly, we considered a geometry corresponding to vertically stacked 2D triangular lattices. The parameters describing, respectively, the hopping within the plane ($t_p = 0.008$ eV) and between dimers ($t_v = 0.08$ eV) were chosen to qualitatively reproduce the main features of the DFT bandstructure. On the other hand, the hopping responsible for the dimerization ($t_D = 0.12$ eV) was chosen to match the experimental $k_z$-dispersion. With these parameters fixed, we tuned the interaction strength to reproduce the optical bandgap, obtaining $U = 0.86$ eV.

### ARPES simulations

The ARPES intensity is calculated with an infinite-layer slab constructed with the bulk DMFT spectral function. The APRES intensity can be written with Fermi's golden rule as:

$$I(\mathbf{k}_\parallel, E) \propto \sum_\alpha |M_\alpha(\mathbf{k}, E)|^2 \delta(\epsilon_\alpha(\mathbf{k}) + \omega - E), \tag{4}$$

where the photoemission matrix element for band $\alpha$ $M_\alpha(\mathbf{k}, E)$ can be represented with the orbital photoemission matrix element $M_j(\mathbf{k}, E)$ with orbital index $j$ and layer index $l$ as:

$$M_\alpha(\mathbf{k}, E) = \sum_{jl} C_{j\alpha}(\mathbf{k}) e^{-i\mathbf{p}\cdot\mathbf{r}_j} e^{z_j/\lambda} \quad (5)$$
$$\times F_\lambda(k_z - p_\perp) M_j(\mathbf{k}, E),$$

where $F_\lambda(q) = \sum_{l=0}^{\infty} e^{ilqc} e^{-cl/\lambda}$, with $c$ the lattice constant and $p_\perp$ the photoelectron momentum component perpendicular to the surface. In the simulation in Fig. 3, $p_\perp = \sqrt{2(E - \Phi)}$ is defined with work function $\Phi = 4$ eV at $\mathbf{k}_\parallel = (0,0)$. $\lambda$ is the photoelectron escape depth. On the other hand, the bulk spectral function in the orbital basis can be expressed as:

$$A_{jj'}(\mathbf{k}, \epsilon) = \sum_\alpha C_{j\alpha}(\mathbf{k}) C_{j'\alpha}^*(\mathbf{k}) \delta(\epsilon - \epsilon_\alpha(\mathbf{k})). \quad (6)$$

Finally, we get:

$$I(\mathbf{k}_\parallel, E) \propto \int dk_z \sum_{jj'} e^{i\mathbf{p}\cdot(\mathbf{r}_j - \mathbf{r}_{j'})} e^{(z_j + z_{j'})/\lambda} \quad (7)$$
$$\times F_\lambda^2(k_z - p_\perp) A_{jj'}(\mathbf{k}, E - \omega) M_j(\mathbf{k}, E) M_{j'}^*(\mathbf{k}, E).$$

In Fig. 3, the ARPES intensity is calculated without the orbital matrix elements ($M_j(\mathbf{k}, E) = 1$). For the photoelectron escape depth, we take the values from the universal curve as a function of the used photon energy. In Supplementary Fig. 6, we discuss the effects of phase modulation in Eq. (7), which leads to the observed ARPES intensity.

## Data availability
All the data used for generating the figures for this work have been deposited in the Open Research Data Repository of the Max Planck Society EDMOND (https://doi.org/10.17617/3.IPYQID).

## Code availability
The MATLAB scripts used to analyze soft X-Ray ARPES data is available at https://github.com/c0deta1ker/ARPESGUI. Other custom scripts used to analyze and plot data are available from the corresponding author N.B.M.S. upon request.

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

## Acknowledgements

We acknowledge beamtime proposals SI29240-1 (Diamond Light Source Ltd.), and 20212108 (Swiss Light Source). J.A.K. acknowledges support by the Swiss National Science Foundation (SNF-Grant No. P500PT_203159). M.A.S. was funded by the European Union (ERC, CAVMAT, project no. 101124492). Y.X. was supported by the National Natural Science Foundation of China (General Program no. 12374163). M.D., J.A.K., E.C.M., B.P., and N.B.M.S. acknowledge Dr. Procopios Christou Constantinou for supporting our beamtime at the Swiss Light Source. N.B.M.S. acknowledges helpful discussions with Tyrel McQueen.

## Author contributions

M.D., J.A.K., B.P., and E.C.M. performed the ARPES experiments under the supervision of N.B.M.S. and with support from M.D.W., V.N.S., S.J., and N.C.P. M.D., M.D.W., and Y.X. performed DFT calculations. M.D. and N.B.M.S. analyzed the experimental data. F.P. and P.W. calculated spectral functions from DMFT, from which Y.Y. and M.S. computed the ARPES intensities. V.H. and C.F. grew the samples. C.K. and G.W. per-formed optical absorption measurements. I.K. performed RBS mea-surements. J.Y., M.N.A., S.S.P.P., and M.A.S. contributed to the interpretation of the data in connection with the JDE. M.D. and N.B.M.S. wrote the manuscript with input from all co-authors. N.B.M.S. conceived and coordinated the project.

## Funding

## Competing interests

The authors declare no competing interests.
