## [Transparent Peer Review file · Nature Communications]

Momentum-Resolved Fingerprint of Mottness in Layer-Dimerized Nb₃Br₈

Corresponding Author: Dr Niels Schröter

Version 0:

Reviewer comments:

Reviewer #1

(Remarks to the Author)

The manuscript by Date et al. represents a simple experiment employing the out-of-plane momentum-dependent photoemission spectra as a novel means to resolve the nature of the insulating state in van der Waals compound Nb₃Br₈. I find the manuscript is generally well laid out and convincing, and the method demonstrated in the manuscript can potentially be used to resolve the Mott vs band insulator in quite a few other related systems. The content and associated impact of the work are within the scope of Nature Communications.

However, there are a few important points related to the manuscript's primary conclusion that require further clarification or refinement:

- 1) Does the rhombohedral crystallographic stacking of Nb₃Br₈ play any role in the k_z dispersion? The primitive unit cell of Nb₃Br₈ contains two Nb₃Br₈ layers, and the authors should clarify how this affects or relates to the band vs Mott insulator scenarios depicted in Fig. 1h and j.
- 2) In comparing the in-plane photoemission with DFT (Fig. 2c,e and d,f), the authors should clarify which set of bands in DFT they are comparing the bands highlighted in the black boxes in Fig. 3e and f to. In DFT, either the set of bands near the Fermi level or the set of bands located at -1eV is close in energy with the highlighted photoemission intensity.
- 3) In Fig.3e, the periodicity of $2\pi/d$ is visible for the blue circles, while for the red circles, the first and second peaks are separated from each other by π/d . This contradicts the same periodicity of red and blue obtained from the DMFT simulation in Fig. 3g for the Mott phase. This discrepancy needs to be discussed and clarified.
- 4) In the conclusion, the authors suggest a pure Mott insulating phase for Nb₃Br₈. I wonder with the current experimental evidence, whether one can completely rule out a cooperative interplay between the Mott and dimerization insulating mechanisms.
- 5) In the photoemission results shown in Fig. 3, the authors should clarify which k_z corresponds to Gamma, and which k_z corresponds to T.

There are a few additional detailed points that should be clarified:

- 1) In Fig. 1e, t_1 and t_2 are not defined anywhere.
- 2) In Fig. 3, how Fig. 3e,g,h are obtained from the first row is not clearly described in words either in text or caption.
- 3) On page 5, the statement "whilst the dimerized band insulator is expected to have a periodicity with half of that lattice constant" is ambiguous. "Lattice constant" on its own often imply that it is the real space lattice constant, which should double in the dimerized phase.

Reviewer #2

(Remarks to the Author)
Journal: Nature Communications

Submission number: NCOMMS-24-66357

Title: Momentum-Resolved Fingerprint of Mottness in Layer-Dimerized Nb₃Br₈

Authors: Mihir Date et al.

Review:

The layer dimerized van der Waals material Nb₃Br₈ has been considered a band insulator for a long time, because it contains an even number of electrons and also because the band structure observed from ARPES is well predicted by DFT. Recently though, this has been put into question because of the observation of a mysterious magnetic field-free diode effect when used as a weak link in Josephson junctions. One possibility that could help explain this phenomenon is if Nb₃Br₈ is instead a dimerized Mott insulator. There was however, prior to this work, no smoking gun that could distinguish between the two types of insulators. This work presents a unique momentum-dependent signature of a Mott insulating phase in Nb₃Br₈. It does so by proposing a novel signature of Mottness in two-dimensional materials which could have impact in many more materials. The work is simple, yet complete and provides important advancements in the detection of Mott physics, but also in understanding Nb₃Br₈ and its field-free Josephson diode effect and potentially time-reversal symmetry breaking superconductivity.

In terms of the methodology, the beauty of this work is that it does not involve overly complicated concepts in order to understand the difference between a dimerized Mott insulator and a dimerized band insulator. Therefore, heavy artillery is not necessary and simple concepts are enough. Still, the authors have supplemented the work with DMFT calculations. They carefully showed that their one-band model was indeed a good representation of the material, as shown in the Supplementary Information section. They obtain their tight-binding parameters based on ARPES fits, which is reasonable. However, that means that the bands are already renormalized by interactions, leading to the question:

- How do these tight-binding parameters change with respect to those extracted from DFT? Can you report the mass renormalization?

Furthermore, the authors claim that a value of 0.86 eV for the on-site Coulomb interaction U “qualitatively reproduces the optical gap”. Still, one could ask:

- What is the value of U predicted by cRPA, as was done, for example, in a first-principles extraction of those parameters in the Ref. 41 of the manuscript?

Let me be clear here: I do not necessarily think performing these calculations simply to answer these questions would add a lot to the manuscript, because as I said, the key concepts promoted in this paper do not require extremely accurate and precise numerical simulations, since the ideas are fairly simple. This could very well be studied in a separate paper that focuses on the predictive power of electronic structure simulations.

Now, perhaps my only suggestion for improvements regards the last panels of Fig. 2, where a new symbol is introduced with not much discussion: B_w . Although there is a short discussion at the end of the “In-plane dispersion of the spectral function”, I do not think it is sufficient and should be revisited. In addition, there are dotted lines in the figure that are, I imagine, supposed to help define this parameter. I found them more confusing than enlightening. I would suggest:

- Explaining a bit more carefully this concept and its implications.

Overall, I think the manuscript is excellent and represents a nice collaboration between theory and experiment. It is well written, clear and straightforward. It shines a new light, not only on a material that is attracting a lot of attention at the moment due to its fascinating emergent phenomenon, but also potentially on many more materials. Therefore I recommend its publication in Nature Communications.

Olivier Gingras

Version 1:

Reviewer comments:

Reviewer #1

(Remarks to the Author)

The authors have satisfactorily addressed the points I have raised about the manuscript in the previous round. With which I would like to recommend publication of the present version of manuscript in Nature Communications.

Author response to comments by Reviewer #1

Reviewer comment: The manuscript by Date et al. represents a simple experiment employing the out-of-plane momentum-dependent photoemission spectra as a novel means to resolve the nature of the insulating state in van der Waals compound Nb₃Br₈. I find the manuscript is generally well laid out and convincing, and the method demonstrated in the manuscript can potentially be used to resolve the Mott vs band insulator in quite a few other related systems. The content and associated impact of the work are within the scope of Nature Communications.

Author response: We thank the reviewer for the positive remarks on the employed methodology, and for considering our work within the scope of Nature Communications.

Reviewer comment: However, there are a few important points related to the manuscript's primary conclusion that require further clarification or refinement:

1) Does the rhombohedral crystallographic stacking of Nb₃Br₈ play any role in the k_z dispersion? The primitive unit cell of Nb₃Br₈ contains two Nb₃Br₈ layers, and the authors should clarify how this affects or relates to the band vs Mott insulator scenarios depicted in Fig. 1h and j.

Author response: The rhombohedral stacking results in inequivalent van der Waals gaps, and as the referee correctly points out, this results in two Nb₃Br₈ layers per unit cell that can be considered as a dimerized bilayer. The degree of dimerization can be quantified by the difference between the hopping constants $\eta = |t_1 - t_2|$, where t_1 describes the intradimer hopping and t_2 the interdimer hopping, respectively, along the out of plane direction (Fig. R1).

Fig.R1: Corrected version of Figure 1d showing the inequivalent hopping between alternating van der Waals gaps due to rhombohedral stacking.

If there were no interlayer hybridization (i.e. $t_1=t_2$) for instance by considering layers of a trigonal lattice that are stacked such that the lattice sites are on top of each other), then we would only expect a single half-filled band dispersing along k_z (since each Nb trimer on each lattice site contributes a single spin-1/2 electron), as shown in figure R2g.

If we now consider a finite dimerization due to the rhombohedral stacking, this leads to a backfolding of the bands and in the case of the band insulator to the opening of the hybridization gap at the Fermi level, as shown in Fig.R2h. Note that since the dimerization is only a small perturbation to the potential along the out-of-plane direction, the backfolded bands in the spectral function will have weak spectral weight away from the crystal momentum where the hybridization appears.

However, when we consider strong onsite interactions together with dimerization, we expect a spectral feature where the Hubbard bands are backfolded and open a hybridization gap away from the Fermi level, as shown in Fig.R2j. Again, the backfolded Hubbard band is weak due to the small perturbation of the dimerization.

The dimerization therefore affects the k_z dispersion in both the band-insulating and the Mott-insulating phases.

Fig.R2: Snippet of Fig.1g-j of the manuscript. Illustration of spectral functions of a (g) metal, (h) dimerized band insulator, (i) Mott insulator, and (j) dimerized Mott insulator.

2) In comparing the in-plane photoemission with DFT (Fig. 2c,e and d,f), the authors should clarify which set of bands in DFT they are comparing the bands highlighted in the black boxes in Fig. 3e and f to. In DFT, either the set of bands near the Fermi level or the set of bands located at -1eV is close in energy with the highlighted photoemission intensity.

Author response: In the discussion of Figure 2, we indeed compare the valence band in DFT to the experimental spectral features near the Fermi level. We have now indicated this by a dashed arrow, as shown below in Fig. R3.

Fig.R3: Modified Fig.2 of the manuscript, improved by incorporating the reviewer's comments.

3) In Fig.3e, the periodicity of $2\pi/d$ is visible for the blue circles, while for the red circles, the first and second peaks are separated from each other by π/d . This contradicts the same periodicity of red and blue obtained from the DMFT simulation in Fig. 3g for the Mott phase. This discrepancy needs to be discussed and clarified.

Author response: We have noticed this discrepancy as well, which appears due an enhancement of the shoulder observed in Fig. R2j for the photon energies in the range of $h\nu=60-65$ eV. At the moment, we have no conclusive explanation for this phenomenon, but we would like to point out that for smaller photon energies, final states in the photoemission process are sometimes not free-electron like. This can lead to deviations of the experimental data from the expected dispersion.

However, the overall dispersion of the spectral function shown in Fig. 3a agrees qualitatively very well with the dispersion predicted for the dimerized Mott insulator (Fig. 3c), and the $2\pi/d$ periodicity of the top of the valence band is also observed over a wider momentum range when probed with soft X-rays (Fig. S4 in supplementary materials), where the final states are expected to be more free-electron like. We therefore conclude that even though not all features of our data can be fully explained, the overall dispersion strongly points towards the dimerized Mott insulating phase in Nb₃Br₈.

We have added the following sentence to our manuscript to discuss this discrepancy

The position of the maxima and minima of the bands can also be extracted from line cuts, so-called momentum distribution curves (MDCs), shown in Fig. 3(e). The blue scatter plot shows MDC intensity peaks, integrated in the energy window around the band maximum, spaced by 0.89 \AA^{-1} . On the other hand, we notice that in the red scatter plot corresponding to the MDC of the band minima show an anomalous peak at $\sim 4.3 \text{ \AA}^{-1}$, which does not seem to obey this periodicity. While we do not have a conclusive explanation for this discrepancy, we point out that such behaviour is only observed for lower photon energies

in the VUV range, where the final state may not be free-electron like. This can lead to deviations in the observed photoemission spectra from the expected one. Nevertheless, the observed ARPES dispersion in Fig.3a agrees qualitatively well with the predicted dispersion for a dimerized Mott insulator in Fig.3c. Furthermore, the $2\pi/d$ periodicity of the band maxima is also observed over a wider momentum range when probed with soft X-rays (see Fig.S4 of Supplementary Information), where the final state can be expected to become more free-electron like. Therefore, we conclude that the overall data strongly points towards the dimerized Mott insulating phase in Nb_3Br_8 .

4) In the conclusion, the authors suggest a pure Mott insulating phase for Nb_3Br_8 . I wonder with the current experimental evidence, whether one can completely rule out a cooperative interplay between the Mott and dimerization insulating mechanisms.

Author response: We thank the reviewer for raising this interesting idea. In our model there is a clear distinction between dimerized-Mott and band-insulating phase. In the former, the self-energy diverges at the Fermi-level, whilst in the latter, the self-energy goes to zero, and the two phases do not co-exist. By fitting our model to the experimentally observed band gap and spectral function, we find that Nb_3Br_8 is deep inside the Mott-phase. Nevertheless, theoretical work in the literature suggests that there can indeed be a coexistence region between the two phases at finite temperature and higher strengths of the Hubbard interaction (*Sentef et al, PRB 80, 155116 (2009)*), which may be explored in future work. We have added a sentence to the conclusions to discuss this possibility:

*Our approach may also be applied to other correlated insulators, such as strongly correlated heterostructures of two-dimensional materials that can be tuned both in terms of screening electronic interactions through dielectric environments and controlling interlayer interactions via deterministic control of stacking and twist angles [34]. A quantum phase transition from band- to Mott-insulator could then be identified by a spectral weight transfer from $kz = \pi/2d$ (band insulator, cf. Fig. 1h) to $kz = \pi/d$ (Mott insulator, cf. Fig. 1j), where d is the average interlayer spacing. **Interestingly, at some combination of hopping strength and onsite correlations, temperature control can even lead to coexisting band and Mott insulating behavior**[*Sentef et al, PRB 80, 155116 (2009)*], which could potentially be explored in such a tunable platform.*

5) In the photoemission results shown in Fig. 3, the authors should clarify which kz corresponds to Gamma, and which kz corresponds to T.

Author response: We thank the reviewer for raising this point. We have now added the kz planes corresponding to the Brillouin zone expected from the rhombohedral crystal structure. Note that due to the weak dimerization and therefore weak spectral intensity of the backfolded bands, the spectral function does not follow the expected periodicity of the dimerized unit cell, but appears to follow a periodicity of an undimerized structure.

Fig.R4: Revised Fig.3 of the manuscript, by incorporating reviewer's comments.

Reviewer comment: There are a few additional detailed points that should be clarified:

1) In Fig. 1e, t_1 and t_2 are not defined anywhere.

Author response: We apologize for the oversight, and thank the reviewer for pointing this out. We have now shown t_1 and t_2 in Fig.1d.

2) In Fig. 3, how Fig. 3e,g,h are obtained from the first row is not clearly described in words either in text or caption.

Author response: We integrated the photoemission intensity in the energy windows shown in red and blue in Fig.3a, c, and d, respectively. We have written this in the caption of Fig.3 as follows:

3) On page 5, the statement "whilst the dimerized band insulator is expected to have a periodicity with half of that lattice constant" is ambiguous. "Lattice constant" on its own often imply that it is the real space lattice constant, which should double in the dimerized phase.

Author response: We apologize for the confusion, and agree with the reviewer. We have made the following changes:

... whilst the dimerized band insulator is expected to have a periodicity with half of that **reciprocal** lattice constant...

Author response to comments by Reviewer #2

The layer dimerized van der Waals material Nb₃Br₈ has been considered a band insulator for a long time, because it contains an even number of electrons and also because the band

structure observed from ARPES is well predicted by DFT. Recently though, this has been put into question because of the observation of a mysterious magnetic field-free diode effect when used as a weak link in Josephson junctions. One possibility that could help explain this phenomenon is if Nb₃Br₈ is instead a dimerized Mott insulator. There was however, prior to this work, no smoking gun that could distinguish between the two types of insulators. This work presents a unique momentum-dependent signature of a Mott insulating phase in Nb₃Br₈. It does so by proposing a novel signature of Mottness in two-dimensional materials which could have impact in many more materials. The work is simple, yet complete and provides important advancements in the detection of Mott physics, but also in understanding Nb₃Br₈ and its field-free Josephson diode effect and potentially time-reversal symmetry breaking superconductivity.

>

> In terms of the methodology, the beauty of this work is that it does not involve overly complicated concepts in order to understand the difference between a dimerized Mott insulator and a dimerized band insulator. Therefore, heavy artillery is not necessary and simple concepts are enough. Still, the authors have supplemented the work with DMFT calculations. They carefully showed that their one-band model was indeed a good representation of the material, as shown in the Supplementary Information section.

Author response: We thank Dr. Gingras for the positive assessment of our results and approach.

They obtain their tight-binding parameters based on ARPES fits, which is reasonable. However, that means that the bands are already renormalized by interactions, leading to the question:

- How do these tight-binding parameters change with respect to those extracted from DFT? Can you report the mass renormalization?

Author response: We thank the Reviewer for his remarks and for the opportunity to improve the presentation of our results. The *ab initio* band structure of this compound is quite complex. Therefore, as Dr. Gingras correctly pointed out, to perform qualitatively relevant calculations, we opted for a minimal model approach. In selecting our tight-binding Hamiltonian, we sought to mimic the DFT band structure (reported in Fig. 2c–2d of the manuscript) in the absence of correlations, which we subsequently incorporated using DMFT. To achieve this, we fine-tuned the in-plane, intra-dimer, and inter-dimer hopping parameters to optimally capture the two bonding/antibonding states near the Fermi level and the optical gap—both in the absence of electron-electron interactions. This procedure yielded a minimal tight-binding model with a gap that was severely underestimated compared to experimental results, necessitating correction through the inclusion of a local self-energy obtained using DMFT.

Regarding the mass renormalization, we cannot provide the Z factor commonly derived within the DMFT framework, as the value of U yielding the optimal gap corresponds to a diverging self-energy. In principle, an estimate could be obtained by comparing the bare DoS to the spectral function from analytical continuation. However, we chose not to follow this route due to the well-known pitfalls of the MaxEnt method at high energy and because the overall bandwidth is widened by the presence of a gap, which shifts the spectral weight to higher energies, complicating a direct comparison with the bare DoS.

Reviewer comment: Furthermore, the authors claim that a value of 0.86 eV for the on-site Coulomb interaction U "qualitatively reproduces the optical gap". Still, one could ask:

- What is the value of U predicted by cRPA, as was done, for example, in a first-principles extraction of those parameters in the Ref. 41 of the manuscript?

Author response: We thank Dr. Gingras for the interesting prompt for reflection. The reference he points out represents a highly valuable study of the similar compound Nb_3Cl_8 .

However, the same paper states that the bulk configuration of this system at high temperature is only mildly affected by inter-dimer hybridizations, and the reported insulating state is attributed to be of the pure Mott type. It is also worth noting that the gap amplitudes found experimentally differ significantly (1.4–2 eV in Nb_3Cl_8 versus 0.9 eV in Nb_3Br_8). Therefore, we believe it would be on the fringe of appropriateness to make a direct comparison between an optimal U value (one that reproduces the optical gap) used in the two contexts. As the referee rightly states, trustworthy statements should be supported by a thorough *ab initio* + cRPA study of the system.

Reviewer comment: Let me be clear here: I do not necessarily think performing these calculations simply to answer these questions would add a lot to the manuscript, because as I said, the key concepts promoted in this paper do not require extremely accurate and precise numerical simulations, since the ideas are fairly simple. This could very well be studied in a separate paper that focuses on the predictive power of electronic structure simulations.

Author response: This is something we will most likely do in the near future.

Reviewer comment: Now, perhaps my only suggestion for improvements regards the last panels of Fig. 2, where a new symbol is introduced with not much discussion: δ_{BW} . Although there is a short discussion at the end of the "In-plane dispersion of the spectral function", I do not think it is sufficient and should be revisited. In addition, there are dotted lines in the figure that are, I imagine, supposed to help define this parameter. I found them more confusing than enlightening. I would suggest:

- Explaining a bit more carefully this concept and its implications.

Author response: δ_{BW} stands for the experimental in-plane 'bandwidth' that we extract from energy dispersive curves (EDCs). We have defined it in the caption of Fig.2, as follows:

... (k) and (l) illustrate the in-plane bandwidth (δ_{BW}) of the VBM in the two kz -planes, respectively, as extracted from the peaks of the EDCs at the Γ -bar point and M -bar point.

Whilst we are currently not making use of these numbers to constrain our theoretical model (which was mainly focused on the kz dispersion), they may become important for future theoretical studies and we therefore report them here without further discussion.

Reviewer comment: Overall, I think the manuscript is excellent and represents a nice collaboration between theory and experiment. It is well written, clear and straightforward. It shines a new light, not only on a material that is attracting a lot of attention at the moment due to its fascinating emergent phenomenon, but also potentially on many more materials. Therefore I recommend its publication in Nature Communications.

Author response: We thank Dr. Gingras for appreciating our collaborative efforts, the manuscript, and for the recommending our work for publication in Nature Communications.